# VRIQ: BENCHMARKING AND ANALYZING VISUAL-REASONING IQ OF VLMS

## ABSTRACT

Recent progress in Vision Language Models (VLMs) has raised the question of whether they can reliably perform nonverbal reasoning. To this end, we introduce VRIQ (Visual Reasoning IQ), a novel benchmark designed to assess and analyze the visual reasoning ability of VLMs. We evaluate models on two sets of tasks: abstract puzzle-style and natural-image reasoning tasks. We find that on abstract puzzles, performance remains near random with an average accuracy of around 28%, while natural tasks yield better but still weak results with 45% accuracy. We also find that tool-augmented reasoning demonstrates only modest improvements. To uncover the source of this weakness, we introduce diagnostic probes targeting perception and reasoning. Our analysis demonstrates that around 56% of failures arise from perception alone, 43% from both perception and reasoning, and only a mere 1% from reasoning alone. This motivates us to design fine-grained diagnostic probe questions targeting specific perception categories (e.g., shape, count, position, 3D/depth), revealing that certain categories cause more failures than others. Our benchmark and analysis establish that current VLMs, even with visual reasoning tools, remain unreliable abstract reasoners, mostly due to perception limitations, and offer a principled basis for improving visual reasoning in multimodal systems.

## 1 INTRODUCTION

Recent years have witnessed remarkable advancements in both Large Language Models (LLMs) and Vision-Language Models (VLMs), transforming how artificial intelligence processes and understands multimodal information. These developments have been driven by novel architectures (Vaswani et al., 2017; Brown et al., 2020; Touvron et al., 2023), enhanced training methodologies (Chung et al., 2024; Ouyang et al., 2022), and massive-scale datasets (Schuhmann et al., 2022; Gadre et al., 2023). Reasoning capabilities have emerged as a cornerstone of LLM performance, with techniques such as chain-of-thought prompting (Wei et al., 2022), self-consistency (Wang et al., 2023a), and tree-of-thoughts (Yao et al., 2023) demonstrating that systematic reasoning dramatically improves performance on complex tasks. These reasoning approaches enable models to decompose problems, maintain logical consistency, and arrive at more accurate conclusions compared to direct answer generation, particularly benefiting mathematical problem-solving, logical deduction, and multi-step planning tasks.

In the visual domain, visual reasoning has become a focus of attention as researchers seek to extend these reasoning benefits to multimodal contexts. Visual reasoning—the ability to analyze, interpret, and draw logical conclusions from visual information—has become increasingly critical given VLMs' deployment in high-stakes applications, including medical diagnosis (Li et al., 2023; Phan et al., 2024), and autonomous navigation (Pan et al., 2024; Chen et al., 2024). Recent efforts have introduced various approaches to enhance visual reasoning in VLMs, including visual chain-of-thought methods (Shao et al., 2024), tool-augmented reasoning systems (Gupta & Kembhavi, 2023; Surís et al., 2023), and reasoning-optimized models like OpenAI's o3 and other specialized architectures (OpenAI, 2025b). However, despite these advances, recent evaluations have revealed persistent limitations in VLMs' ability to perform complex visual reasoning tasks (Fu et al., 2024c; Lu et al., 2024b; Yu et al., 2023), particularly when abstract thinking and multi-step inference are required.

To address these challenges, it is crucial to systematically evaluate current VLMs' failure modes and identify specific areas requiring improvement. Several benchmarks have been proposed to assess visual reasoning capabilities (Li et al., 2024a; Jiang et al., 2024; Zhang et al., 2025a), yet they exhibit notable limitations. Some focus exclusively on natural image questions where reasoning requirements are relatively shallow and single-step (Fu et al., 2024a; Ging et al., 2024), while others concentrate solely on abstract reasoning tasks such as IQ-style puzzles without connecting to real-world visual understanding (Zhang et al., 2019; Barrett et al., 2018). Furthermore, existing benchmarks typically only evaluate perception and reasoning as monolithic capabilities (Cai et al., 2025a). Here, perception refers to the ability to accurately identify visual elements (e.g., shapes, counts, positions), while reasoning is the subsequent process of applying abstract rules and principles to that perceived information to solve problems. Critically, no existing benchmark comprehensively evaluates: (1) how models perform on both natural and abstract visual questions using identical logical reasoning methodologies (such as odd-one-out problems), allowing direct comparison of model behavior across these domains, and (2) fine-grained diagnostic analysis identifying exact failure modes in specific perceptual and reasoning sub-capabilities, including color recognition, 3D/depth perception, counting, spatial positioning, rotation detection, and reasoning from textual descriptions versus actual images.

To investigate this gap, we present a comprehensive benchmark for evaluating visual reasoning in VLMs. Our benchmark comprises two parallel sets of questions—abstract image reasoning and natural image reasoning—organized under identical IQ test categories to enable direct comparison of model performance across visual domains. Questions are specifically designed to challenge VLMs' capabilities across multiple dimensions. Additionally, we develop diagnostic probe questions for each IQ test, with each probe targeting specific aspects of perception or reasoning required to solve the main task. These probes, created through human annotation, enable us to dissect model failures systematically. Our analysis reveals three key insights: (i) both open source and proprietary models perform badly on the benchmark, even though it consists mostly of elementary school-grade IQ questions, see subsection 6.1, (ii) we quantify the distribution of errors across perception-only, reasoning-only, and combined perception-reasoning failures, and find that perception limitations are the main culprit, see subsection 6.2, and (iii) we measure the effect of different perception categories (counting, detail recognition, color identification, position understanding, 3D/depth perception, etc.) on the visual reasoning abilities of VLMs and find that it differs sizably, see subsection 6.3. This benchmark and evaluation methodology for decomposing visual reasoning tasks facilitates precise failure detection and provides actionable insights for developing more capable models in both abstract and natural image reasoning. In summary, our contributions are:

1. A novel multidimensional visual reasoning benchmark consisting of five main IQ question categories containing both abstract and natural reasoning questions, enabling direct comparison of model performance across visual domains.

2. A hierarchical evaluation framework incorporating specific perception and reasoning probe questions to enable fine-grained diagnosis of model capabilities across sub-perceptual and reasoning tasks, revealing precise failure modes.

3. Extensive experiments on a wide range of state-of-the-art VLMs, providing detailed insights into their strengths, weaknesses, and specific areas requiring improvement for advancing visual reasoning capabilities.

We plan to release the complete benchmark, all corresponding probing questions, and the methodology for constructing diagnostic probes for given images, facilitating community efforts toward developing more capable and reliable visual reasoning systems.

## 2 RELATED WORK

**Multimodal benchmarks.** As the multimodal foundation model made significant progress in traditional vision tasks, it becomes increasingly important to develop an effective way of measuring their success. To achieve this, many general-purpose Visual Question Answering (VQA) tasks (Mathew et al., 2021; Gurari et al., 2018; Antol et al., 2015; Goyal et al., 2017; Singh et al., 2019; Hudson & Manning, 2019; Mathew et al., 2021; Marino et al., 2019; Li et al., 2024b; Liu et al., 2023; Liang et al., 2024; Saikh & et al., 2022; Yue et al., 2024b; Wang et al., 2023b; Chen et al., 2023)

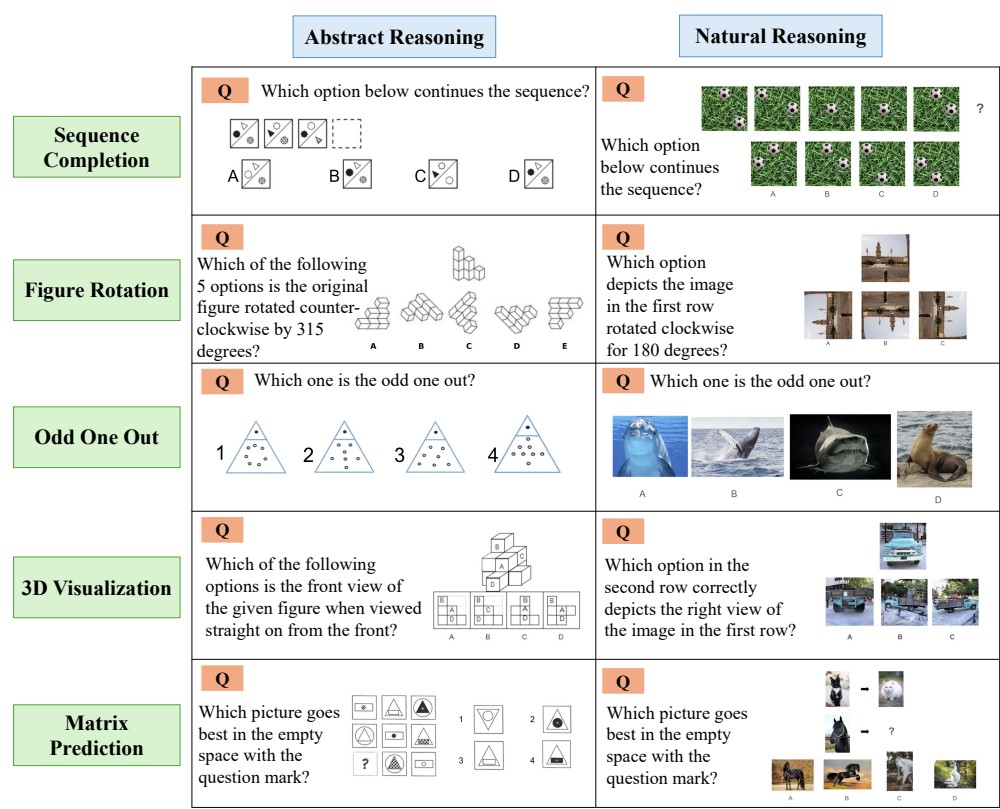

Figure 1: Sampled VRIQ examples from each question type across abstract and natural categories.

have been proposed. To provide a more focused evaluation for, many specialized benchmarks have been introduced: (Lu et al., 2024a; Zhang et al., 2024a; Qiao et al., 2024; Wang et al., 2024d;a; Li et al., 2024c; Lu et al., 2021; Li et al., 2025b; Wang et al., 2024c; Doris et al., 2024). Ours difference from them in that we specialize in puzzle and IQ problem solving and focus on analysis.

**Solving multimodal IQ problems.** Efforts have been made to evaluate MLLMs on solving puzzle or IQ problems. Zhang et al. (2025b) focus on evaluating simple puzzle solving without symbolic or IQ questions. Li et al. (2025a) introduces more complex puzzle problem. MMIQ (Cai et al., 2025b) is the closest to ours, which also evaluates multimodal performance on IQ problems; however, unlike MMIQ, our benchmark introduces parallel abstract–natural puzzle families and a hierarchical probing framework, enabling fine-grained attribution of failures to perceptual versus reasoning sources.

## 3 VRIQ: A DIAGNOSTIC VISUAL REASONING BENCHMARK

We introduce **VRIQ (Visual Reasoning IQ)**, a diagnostic benchmark of *N=440* expert-authored items that evaluates vision–language models on both *perception* and *reasoning* using IQ-style puzzles with multi-dimensional annotations. Unlike algorithmically generated sets (e.g., PGM, RAVEN), VRIQ provides hand-crafted items with **eight perceptual** and **five reasoning** tags, enabling reproducible error attribution and failure-mode analysis.

### 3.1 BENCHMARK DESIGN PRINCIPLES

**Parallel Domain Construction.** VRIQ features two puzzle families—abstract and natural—that share the same reasoning category and logical structure while differing in visual domain. Each reasoning category (e.g., Sequence Completion, Matrix Prediction) is instantiated in both domains,

Table 1: VRIQ Question distribution across categories and domains.

| IQ Test Category | Abstract Questions | Natural Questions |
|---|---|---|
| Sequence Completion | 50 | 20 |
| 3D Reasoning | 90 | 20 |
| Figure Rotation | 100 | 20 |
| Odd One Out | 50 | 20 |
| Matrix Prediction | 50 | 20 |

allowing controlled comparison of abstract reasoning versus semantics-grounded inference. Figure 2 illustrates representative paired examples across our five reasoning categories.

- **Abstract puzzles** were designed by drawing inspiration from publicly available abstract reasoning questions. They employ geometric primitives (shapes, lines, patterns) with multi-attribute transformations and high-similarity distractors. Unlike procedurally generated datasets that vary single attributes, our hand-crafted instances require tracking multiple simultaneous transformations, providing a more rigorous test of compositional understanding.

- **Natural puzzles** were designed to preserve the same category-specific reasoning requirements while using real-world objects and scenes. For instance, an Odd One Out task may require identifying functional differences among objects rather than geometric properties. Natural images were either sourced from open repositories or generated using state-of-the-art vision–language models (GPT-4V (OpenAI, 2025a), Gemini Pro Vision (Team et al., 2023)). All images were manually validated to ensure logical consistency and to filter out generative artifacts (see Appendix B for samples).

**Coverage of Cognitive Primitives.** VRIQ spans five fundamental reasoning categories established in human psychometrics: (1) Sequence Completion (SC), (2) Matrix Prediction (MP), (3) Odd One Out (OO), (4) Figure Rotation (FR), and (5) 3D Visualization (3D). Table 1 presents the distribution of 440 puzzles across categories and domains. Each category includes both abstract and natural variants to enable controlled comparison within the same reasoning type.

## 4 EVALUATION FRAMEWORK

Our evaluation framework employs a three-tier diagnostic hierarchy to decompose vision–language model failures, enabling attribution of errors to perception, reasoning, or their interaction.

### 4.1 TIER 1: END-TO-END ACCURACY

We first measure overall puzzle accuracy across the five reasoning categories of VRIQ: Sequence Completion (SC), Matrix Prediction (MP), Odd One Out (OO), Figure Rotation (FR), and 3D Visualization (3D). Each category includes both abstract and natural variants, allowing us to compare performance in settings that demand symbolic reasoning versus those grounded in semantics. This tier reflects combined perceptual and reasoning competence.

### 4.2 TIER 2: DIAGNOSTIC PROBES

To analyze errors, we design two complementary probe types: (i) **Perceptual Probes:** Atomic questions that isolate a single visual attribute (e.g., *"How many red circles are shown?"*). These test visual extraction directly without requiring reasoning, and (ii) **Reasoning Probes:** Text-only reformulations of each puzzle that preserve the logical structure while removing visual complexity. For example, a visual matrix with $45°$ rotations is rewritten as: *"If each element rotates clockwise by $45°$, and position 3 shows $90°$ rotation, what appears at position 4?"*. These test logical inference given explicit perceptual facts.

Comparing probe outcomes to end-to-end accuracy allows us to identify whether a failure stems primarily from perception, reasoning, or both.

## 4.3 TIER 3: ERROR CATEGORIZATION

Using probe performance, we classify errors into three categories. **Perception-only** errors occur when the model fails perceptual probes but succeeds on reasoning probes, indicating difficulty in extracting visual features despite intact reasoning. **Reasoning-only** errors arise when the model passes perceptual probes but fails reasoning probes, showing that while perception is correct, logical inference goes wrong. Finally, **Combined** errors appear when the model fails both probe types, reflecting cascading failures across perception and reasoning. This categorization distinguishes independent reasoning deficits from those that arise due to perceptual mistakes.

## 4.4 MULTI-DIMENSIONAL ANNOTATION SCHEMA

Each puzzle is annotated along eight perceptual dimensions to ensure systematic coverage: Color, Shape, Count, Position, Rotation/Orientation, 3D/Depth, Symmetry/Pattern, and Distractor Similarity. These dimensions were chosen based on recurring failure modes of VLMs documented in prior work (Rahmanzadehgervi et al., 2024; Kamath et al., 2023; Zhang et al., 2024b; Khemlani et al., 2025). Annotations ensure that each puzzle's perceptual demands are explicit, and they enable alignment between probe design and puzzle requirements.

## 4.5 PROBE CONSTRUCTION PROTOCOL

For each puzzle, we generate probes corresponding only to the perceptual dimensions required to solve it. This avoids introducing irrelevant attributes and ensures diagnostic precision. Probes are designed to be trivial for humans, so any model failure can be unambiguously attributed to the intended dimension. See Appendix A for a detailed worked example of a matrix reasoning puzzle decomposed into perceptual and reasoning probes. This hierarchical design moves beyond aggregate accuracy to reveal whether failures are caused by perception, reasoning, or their interaction, providing actionable signals for model analysis and improvement.

## 5 EXPERIMENTAL SETUP

We evaluate a diverse set of open-source and proprietary vision–language models (VLMs), chosen to span model scale, architecture, and reasoning capabilities:

**Open-source models** Qwen2.5-VL-3B-Instruct (Huggingface, 2025b), Qwen2.5-VL-3B-Instruct-AWQ (quantized) (Huggingface, 2025c), Qwen2.5-VL-7B-Instruct (Huggingface, 2025d), InternVL3-9B (Huggingface, 2025a), llava-v1.6-mistral-7b-hf (Huggingface, 2025e), llava-v1.6-vicuna-7b-hf (Huggingface, 2025f). These models represent the widely used open community baselines, covering both instruction-tuned and quantized variants. We include them to assess how parameter count and efficiency trade-offs affect both perceptual and reasoning probes.

**Frontier / proprietary models** GPT-4o (AI, 2025a) , GPT-4o-mini (AI, 2025b), OpenAI o3 ("thinking with images") OpenAI (2025b). These API-served models are larger and trained with broader multimodal pretraining pipelines. GPT-4o and GPT-4o-mini are optimized for image-text interaction, while o3 is explicitly designed for reasoning with tool use.

**Special Case: o3 and Tool-Enhanced Reasoning.** Unlike conventional VLMs, OpenAI o3 incorporates native tool use into its reasoning pipeline. Beyond static perception, o3 can autonomously perform operations such as cropping, zooming, and rotating images, while also invoking code execution, external retrieval, and image generation when needed. This integration allows the model to think with images—not only interpreting them but actively manipulating visual inputs during problem solving. As a result, o3 exhibits more agentic reasoning loops, where perception and action co-evolve to refine intermediate representations and solutions OpenAI (2025c).

We include o3 in our evaluation to test whether such tool-augmented visual reasoning leads to measurable gains over static perception mode ls, particularly in benchmarks where fine-grained inspection, transformation, or iterative reasoning over visual inputs is required. This provides insight into how tool-enhanced multimodal systems can extend beyond traditional VLM capabilities.

Table 2: Accuracy (%) of different vision–language models (VLMs) on the VRIQ benchmark across six categories: SC = Sequence Completion, 3D V = 3D Visualization (Three Views), 3D F = 3D Folding/Expansion, FR = Figure Rotation, OO = Odd One Out, and MP = Matrix Prediction. We classify 3D reasoning into two parts (3D V and 3D F) when reporting accuracy.

| Model | SC (%) | 3D V (%) | 3D F (%) | FR (%) | OO (%) | MP (%) | Avg (%) |
|---|---|---|---|---|---|---|---|
| Random Guess | 25 | 20 | 25 | 25 | 25 | 25 | 24.17 |
| **Open Source VLMs** | | | | | | | |
| Qwen2.5-VL-3B-Instruct | 10.81 | 38.46 | 16 | 25 | 33.33 | 20.69 | 24.05 |
| Qwen2.5-VL-3B-Instruct-AWQ | 24.32 | 38.46 | 20 | 23 | 60 | 34.48 | 33.38 |
| Qwen2.5-VL-7B-Instruct | 40.54 | 30.77 | 36 | 25 | 26.67 | 31.03 | 31.67 |
| InternVL3-9B | 24.32 | 32.31 | 32 | 20 | 16.67 | 31.03 | 26.05 |
| llava-v1.6-mistral-7b-hf | 21.62 | 29.23 | 40 | 20 | 30 | 20.69 | 26.92 |
| llava-v1.6-vicuna-7b-hf | 13.51 | 24.62 | 36 | 26 | 0 | 34.48 | 22.44 |
| **Proprietary VLMs** | | | | | | | |
| gpt-4o | 24.32 | 35.38 | 28 | 28 | 16.67 | 31.03 | 27.23 |
| gpt-4o-mini | 24.32 | 33.85 | 20 | 16 | 16 | 31.03 | 23.53 |
| gpt-o3 (thinking with image) | 48.65 | 38.46 | 36 | 66 | 50 | 41.38 | 46.75 |
| Avg % | 25.74 | 32.15 | 28.90 | 27.40 | 27.43 | 30.08 | 28.62 |

Table 3: Accuracy on natural reasoning categories.

| Model | SC (%) | OO (%) | FR (%) | 3D (%) | MP (%) | Avg (%) |
|---|---|---|---|---|---|---|
| Random Guess | 25 | 25 | 33 | 33 | 25 | 28.2 |
| **Open Source VLMs** | | | | | | |
| Qwen2.5-VL-3B-Instruct | 20 | 60 | 40 | 50 | 0 | 34 |
| Qwen2.5-VL-3B-Instruct-AWQ | 40 | 60 | 35 | 40 | 40 | 43 |
| Qwen2.5-VL-7B-Instruct | 25 | 50 | 35 | 0 | 40 | 30 |
| InternVL3-9B | 40 | 40 | 45 | 40 | 35 | 40 |
| llava-v1.6-mistral-7b-hf | 40 | 10 | 50 | 60 | 45 | 41 |
| llava-v1.6-vicuna-7b-hf | 15 | 50 | 15 | 60 | 15 | 31 |
| **Proprietary VLMs** | | | | | | |
| gpt-4o | 50 | 50 | 40 | 80 | 80 | 60 |
| gpt-4o-mini | 40 | 40 | 20 | 60 | 50 | 42 |
| gpt-o3 (thinking with image) | 80 | 70 | 100 | 100 | 75 | 85 |
| Avg % | 38.89 | 47.78 | 42.22 | 54.44 | 42.22 | 45.11 |

## 6 RESULTS

In this section, we summarize our key results from evaluation on VRIQ using the models mentioned in section 5. The results for abstract questions are displayed in Table 2 and their natural variants are shown in Table 3. Unless otherwise mentioned, we report the average accuracy in % across three random seeded runs for each cell entry. We also include a random guess baseline for reference.

### 6.1 MAIN RESULTS

**MLLMs fail at abstract IQ questions–even proprietary ones.** We observed that across all 6 categories, MLLMs show overall poor performance. Among all models tested [1], Qwen2.5-VL-3B-Instruct-AWQ (Bai et al., 2025) shows the best performance, scoring an average of 33.38% across all categories, although it is only 9.21% above the random guess baseline. Overall, when averaged among all MLLMs tested, the improvement compared to random guess is less than 4% on average despite these problems being easily solvable by humans. Surprisingly, our results show that **even proprietary VLMs do not show much better performance** with abstract IQ questions. For example, gpt-4o (Hurst & OpenAI, 2024) and gpt-4o-mini show even worse performance compared to Qwen2.5-VL-7B models despite being much larger. This observation diverges from prior work

---

[1]Note that this excludes gpt-o3 with "thinking with image" as it's in a different model family can leverage external tools. We further analyze the details of its improvement in section 7

Table 4: Perception vs. reasoning error distribution. Percentage of failures attributed to perception-only, reasoning-only, and combined failures across five VLMs.

| Model | Perception (%) | Reasoning (%) | Both (%) |
|---|---|---|---|
| ChatGPT 4o mini | 56.36 | 0.00 | 43.64 |
| ChatGPT 4o | 60.00 | 3.64 | 36.36 |
| Qwen-3B-AWQ | 56.36 | 0.00 | 43.64 |
| Qwen3B | 52.73 | 1.82 | 43.64 |
| Qwen-7B | 52.73 | 0.00 | 47.27 |
| Avg (%) | 55.84 | 1.10 | 43.07 |

Table 5: Detailed perception probe failure rates. Failure rates (%) for each probe type across IQ categories, with averages reported in the final column. N/A indicates probes not applicable to a given task category.

| Probe Type | SC (%) | MP (%) | OO (%) | FR (%) | 3D (%) | Avg (%) |
|---|---|---|---|---|---|---|
| **Color** | 26.67 | 47.50 | 20.00 | N/A | N/A | 31.39 |
| **Shape** | 35.68 | 60.00 | 65.00 | 80.00 | N/A | 60.17 |
| **Count** | 64.29 | 80.00 | 77.50 | 75.00 | 72.21 | 73.80 |
| **Position** | 56.22 | 73.33 | 50.00 | 51.43 | 60.00 | 58.20 |
| **Rotation/Orientation** | 45.39 | 50.00 | 50.00 | 64.00 | N/A | 52.35 |
| **3D/Depth** | N/A | N/A | N/A | 70.00 | 65.00 | 67.5 |
| **Symmetry/Pattern** | 40.91 | 46.67 | 75.00 | N/A | N/A | 54.19 |
| **Distractor Similarity** | 52.00 | N/A | 46.33 | N/A | N/A | 49.17 |

which finds proprietary MLLMs to perform considerably better than open-sourced ones (Yue et al., 2024a; 2025; Fu et al., 2024b; Li et al., 2025b; Wang et al., 2024b; Lu et al., 2024c). **This suggests that scaling model sizes and training data is not enough to address the perception limitations of models** as we discuss in detail in subsection 6.2. More importantly, we note that the abstract IQ questions in our benchmark can all be found in a myriad of decades-old IQ tests regularly used to assess human IQ across a wide range of ages; failure on these problems reveals a particular and critical deficiency in tackling abstract symbolic problems even with state-of-the-art multimodal foundation models. We conduct in-depth analysis over the errors in section 7 and show that surprisingly, most of the errors come from perception rather than reasoning.

**Different models exhibit different strengths.** Although the overall accuracy of each model is relatively poor, closer inspection reveals substantial variance across categories (Table 2, Table 3). This variance indicates that current MLLMs do not fail uniformly but instead display heterogeneous strengths and weaknesses. For example, the best-performing model on matrix prediction (MP), `llava-v1.6-vicuna-7b-hf` (Liu et al., 2024), achieves $34.5\%$, yet it collapses to $0\%$ on odd-one-out (OO). Conversely, `Qwen2.5-VL-3B-AWQ` achieves a striking $60\%$ on OO but remains close to random guess (20–34%) in other categories. This cross-category divergence is not limited to open-source models: surprisingly, the `gpt-4o` family, despite its scale and proprietary training, does not dominate in any single category and often trails behind smaller open-source alternatives.

Such patterns underscore an important observation: *while no model excels universally, each exhibits localized competence in specific problem types*. This specialization suggests that architectural or training differences—such as alignment objectives, quantization, or reliance on language priors—may selectively benefit certain categories while hindering others. The practical implication is that model choice should depend strongly on the target task distribution, as relying on a single "state-of-the-art" model may be insufficient for broad-spectrum reasoning.

**MLLMs perform better on natural variants, but remain unreliable.** The natural variants of our IQ benchmark reveal a markedly different trend compared to their abstract counterparts (Table 3). Whereas abstract reasoning tasks reduce most models to near-chance accuracy, grounding the problems in natural imagery leads to consistent gains of 10–20 percentage points above random

guess across nearly all categories. For example, average performance on sequence completion improves from 25.8% in the abstract setting to 38.9% in the natural setting, and on Odd One Out from 27.4% to 47.8%. These improvements demonstrate that perceptual grounding substantially reduces task difficulty. This is reasonable since these models are mainly trained on natural image based QAs instead of symbolic ones. Crucially, it supports our hypothesis that many errors observed in abstract settings arise not purely from reasoning deficits but from failures of low-level visual perception.

Despite this encouraging trend, reliability remains limited. Even in the natural setting, performance is far from human-level: the best open-source models cluster around 35–45%, while proprietary models like GPT-4o reach 80% on 3D tasks but hover near 50% on others. Moreover, variance across categories and models is striking. For instance, `llava-vicuna-7b` attains 60% on 3D reasoning yet only 15% on sequence completion, reflecting an inconsistency that precludes dependable deployment. This heterogeneity suggests that gains from natural grounding are unevenly distributed across categories and architectures, indicating that current models do not acquire a generalizable "IQ-like" competence but rather exploit superficial cues available in specific tasks. We provide more detailed studies in subsection 6.3.

**Using tools to "think with image" largely improves performance.** Across both abstract and natural variants, although most models show poor performance, we observe a substantial leap with `gpt-o3` (tool-augmented) (OpenAI, 2025). Unlike other `gpt` series models such as `gpt-4o` or `gpt-4o-mini`, `gpt-o3` is equipped with external tool use, allowing the model to apply Python-coded operations such as "cropping," "magnification," and "distortion." These operations enable the model to actively manipulate the input image, thereby enhancing visual perception by focusing on the most relevant regions and reducing hallucinations. On the abstract benchmark, `gpt-o3` achieves 48.7% on sequence completion and 66% on rotation, compared to near-random performance ($\sim 25\%$) by all other models. On the natural benchmark, the gains are even more striking: it reaches 80% on sequence completion, 70% on odd-one-out, and 100% on both spatial reasoning and 3D interpretation—often more than doubling the accuracy of the strongest baseline. This makes `gpt-o3` the only model in our study to approach human-like competence on certain tasks.

These results underscore two broader implications. First, the bottleneck in MLLMs on IQ-style reasoning is not only architectural but also methodological: augmenting models with interactive perception tools enables qualitatively different behaviors that static end-to-end training has not yet achieved. Second, while `gpt-o3` demonstrates the promise of tool-augmented reasoning, it also highlights a growing divergence between models designed as standalone systems and those integrated into richer tool-use ecosystems. This suggests that **intricate multimodal reasoning may depend less on scaling single models but more on enabling them to orchestrate external procedures that compensate for their perceptual and reasoning blind spots.**

## 6.2 PERCEPTION VERSUS REASONING

To understand the root causes behind poor performance on abstract IQ tasks, we conducted fine-grained diagnostic analysis using our hierarchical probe methodology on five representative models: ChatGPT-4o-mini, ChatGPT-4o, Qwen-3B-AWQ, QwenB, and Qwen-7B. Human annotators classified each failure as perception-only, reasoning-only, or combined perception-reasoning error. Table 4 reveals a striking pattern: perception errors dominate across all models, accounting for 55.4% of failures on average, while reasoning-only errors represent just 1.1%—a 50:1 ratio that holds remarkably consistent regardless of model scale or architecture.

The near-absence of pure reasoning failures (0-3.64% across models) indicates that when visual elements are correctly perceived, models reliably apply appropriate logical operations. The substantial "Both" category (36.4-47.3%) likely represents perception errors cascading into apparent reasoning mistakes rather than independent dual failures, given the rarity of isolated reasoning errors.

Key finding: These results fundamentally challenge assumptions about abstract visual reasoning requirements. **Rather than needing more sophisticated reasoning mechanisms, models face an almost entirely perceptual bottleneck—struggling to extract visual information accurately** rather than to apply logical operations to that information.

## 6.3 FINE-GRAINED PROBE ANALYSIS.

To diagnose perception bottlenecks in more detail, Table 5 reports probe-level accuracy across categories, averaged over five VLMs. Some probe types are marked N/A, as they were not relevant for specific IQ categories (e.g., 3D/Depth was not designed into Sequence Completion). This confirms that our diagnostic design is targeted: probes reflect only the perceptual attributes actually needed for solving each category.

A clear pattern emerges: certain perceptual categories exhibit consistently high failure rates. Counting (73.8%), position (58.2%), and rotation/orientation (52.4%) show systematic weaknesses across models and categories. These attributes require precise spatial encoding and robust relational perception, where even minor visual noise cascades into reasoning errors. By contrast, color shows substantially lower failure rates (31.4%), suggesting that models' visual backbones are better tuned for appearance-based attributes than for spatial or quantitative ones—likely reflecting dataset biases toward object-level recognition rather than relational structure.

Errors in 3D/Depth (67.5%) and symmetry/pattern (54.2%) further demonstrate that VLMs lack strong 3D geometric priors and abstract relational sensitivity. Together, these probe results confirm that perception failures are not monolithic, but span multiple subtypes with distinct difficulty profiles, underscoring the need for perception-focused improvements to unlock more reliable reasoning.

## 7 DISCUSSION AND LIMITATIONS

**Implications for Model Development.** Together, these findings suggest that advancing VLM reasoning requires targeted improvements to perceptual grounding rather than simply scaling reasoning capacity. Enhancing spatial and quantitative perception—through stronger geometric priors, explicit counting mechanisms, or training data emphasizing relational structure—may yield more reliable reasoning than purely architectural tweaks to attention or prompting. Furthermore, evaluation benchmarks should explicitly disentangle perception from reasoning to avoid conflating perceptual noise with logical inference.

**Limitations.** While VRIQ provides a principled diagnostic framework for visual reasoning, our focus on static IQ-style puzzles necessarily limits scope. Dynamic reasoning tasks—temporal prediction, causal inference in video, or embodied navigation—may reveal different failure patterns. Additionally, our probe set targets core visual primitives (count, position, rotation); extending to higher-level attributes (texture, affordances, material properties) could uncover additional bottlenecks. Future work should test whether the perception-reasoning hierarchy we identify generalizes to richer domains like chart understanding, scientific diagrams, and multimodal documents. Despite these, our findings that perception constitutes the dominant bottleneck while pure reasoning failures remain rare suggests a fundamental constraint that likely extends beyond our specific puzzle domain.

## 8 CONCLUSION

We introduced VRIQ, a diagnostic benchmark that decomposes visual reasoning failures into perceptual and reasoning components through targeted probes that systematically isolate these capabilities. Our analysis shows that perception is the primary challenge across representative VLMs: perception-only errors account for most failures, while reasoning-only errors are rare. Certain dimensions, particularly counting, spatial localization, and orientation, exhibit consistently high error rates, whereas attributes like color are handled more reliably, suggesting current encoders favor appearance over structural or quantitative understanding.

These findings indicate that advancing visual reasoning requires prioritizing perceptual robustness in quantitative and spatial domains rather than focusing solely on reasoning architectures. By decomposing performance into interpretable components, VRIQ shifts evaluation from descriptive accuracy to diagnostic insight, providing a principled framework for developing VLMs with stronger perceptual and reasoning foundations.

ETHICS STATEMENT

This work adheres to the ICLR Code of Ethics.[2] Our study does not involve human subjects, private data, or personally identifiable information. The VRIQ benchmark is constructed from synthetic and publicly available image sources. All natural puzzles generated with foundation models were manually reviewed to filter out artifacts and ensure logical consistency. We release the dataset solely for research purposes, with the goal of advancing transparency in the evaluation of multimodal models. We do not anticipate direct harmful applications, but acknowledge that benchmarks can be misused as over-optimistic performance indicators. To mitigate this, we emphasize the diagnostic nature of VRIQ and caution against equating benchmark scores with general reasoning competence.

REPRODUCIBILITY STATEMENT

We have made every effort to ensure reproducibility of our results. Section 5 details the experimental setup, including model versions, prompting formats, decoding parameters, and evaluation metrics. Appendix A provides a worked example of the diagnostic probe methodology. The full benchmark, probe annotations, and evaluation scripts will be released. This release is intended to provide the community with a transparent and standardized resource for diagnosing perception and reasoning capabilities in multimodal models.

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

## A    DIAGNOSTIC PROBE EXAMPLE

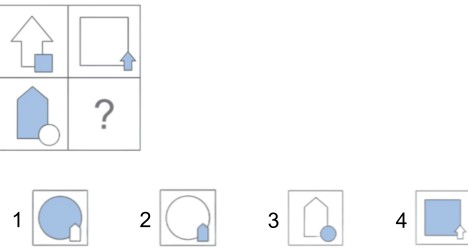

Figure 2: Sample VRIQ example with corresponsing probe questions

This example demonstrates our hierarchical annotation framework applied to a $2 \times 2$ matrix reasoning puzzle. We design targeted probes that systematically isolate different cognitive capabilities required to solve this puzzle.

PERCEPTUAL PROBES

These verify accurate perception of individual visual elements without requiring pattern recognition:

**Shape Dimension:**

- **Q:** "In the top-left panel, what is the small shape located inside the larger shape?"
    - → Expected: *square*
- **Q:** "In the bottom-left panel, what is the small shape?"
    - → Expected: *circle*

**Position Dimension:**

- **Q:** "In the top-right panel, is the arrow the large or the small shape?"
    - → Expected: *small*
- **Q:** "In option 2, is the circle the large shape or the small shape?"
    - → Expected: *large*

**Detail Dimension:**

- **Q:** "In the top-left panel, is the large shape a solid color or an outline?"
    - → Expected: *outline*
- **Q:** "In the bottom-left panel, is the large shape a solid color or an outline?"
    - → Expected: *solid*

REASONING PROBES

These provide complete visual information as text, isolating logical inference capabilities:

- **Q:** "Given a matrix where top-left has a large outline arrow with small filled square, top-right has large outline rectangle with small filled arrow, and bottom-left has large filled pentagon with small outline circle—would bottom-right have a small outline pentagon?"

    → Expected: *yes*

- **Q:** "Given the same matrix setup, would bottom-right have a large outline circle?"

    → Expected: *no*

## B  NATURAL DATASET SAMPLES

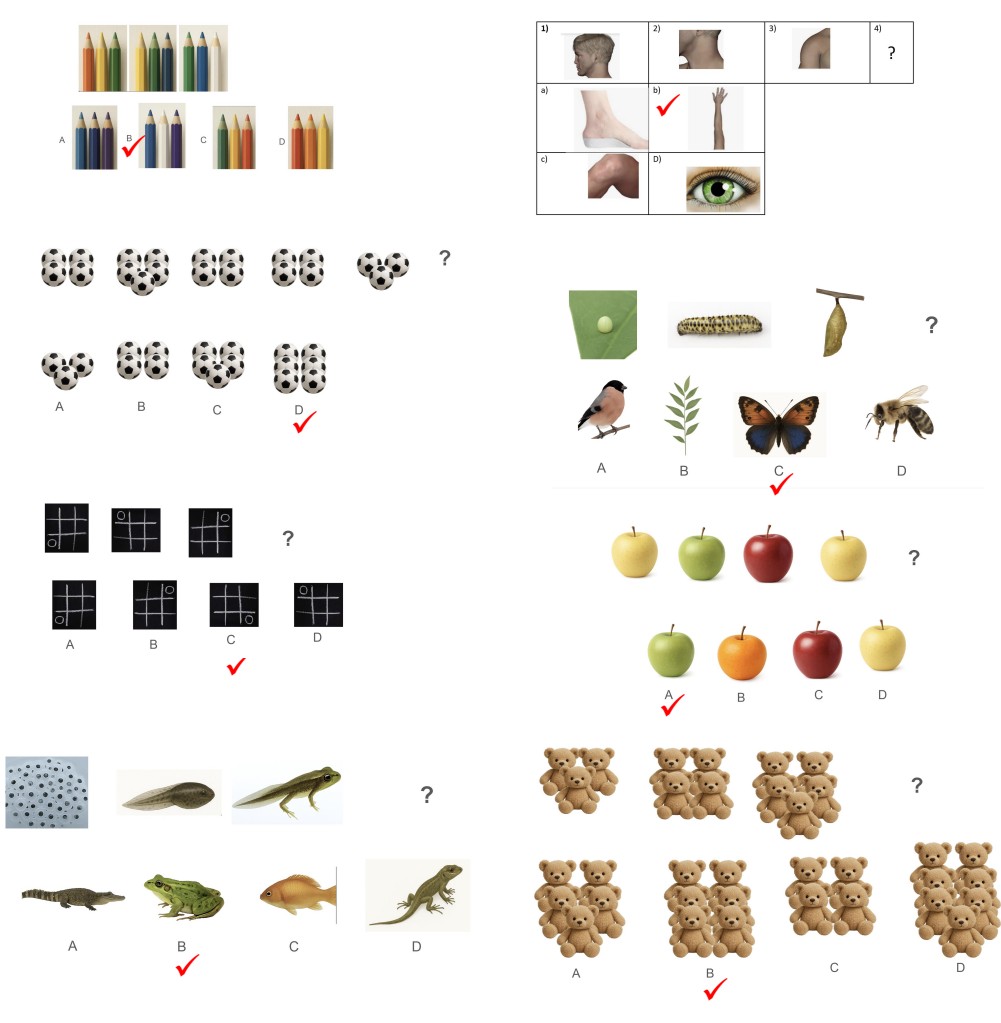

Figure 3: Sample VRIQ natural Sequence Completion questions

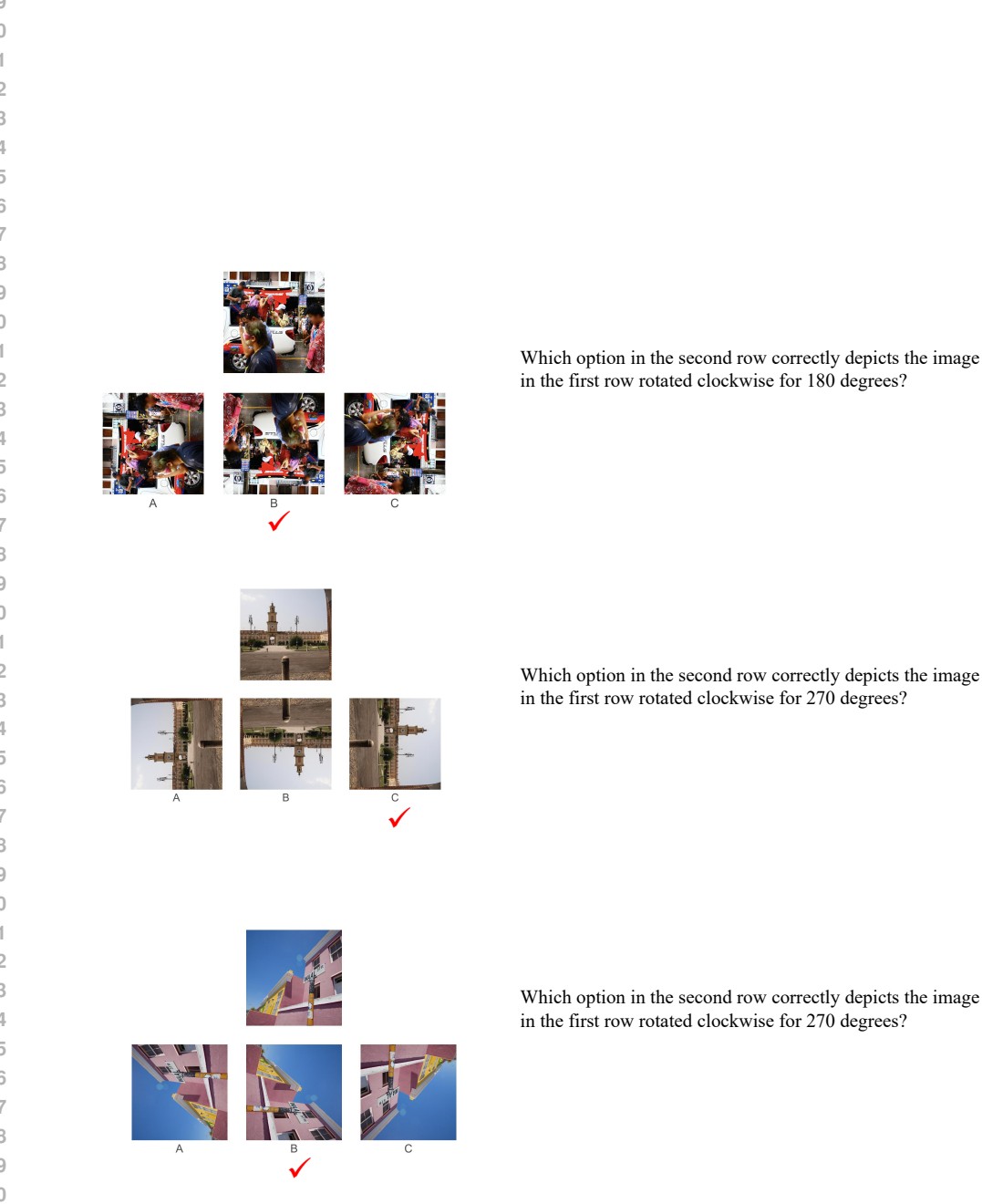

Which option in the second row correctly depicts the image in the first row rotated clockwise for 180 degrees?

Which option in the second row correctly depicts the image in the first row rotated clockwise for 270 degrees?

Which option in the second row correctly depicts the image in the first row rotated clockwise for 270 degrees?

Figure 4: Sample VRIQ natural Rotation questions

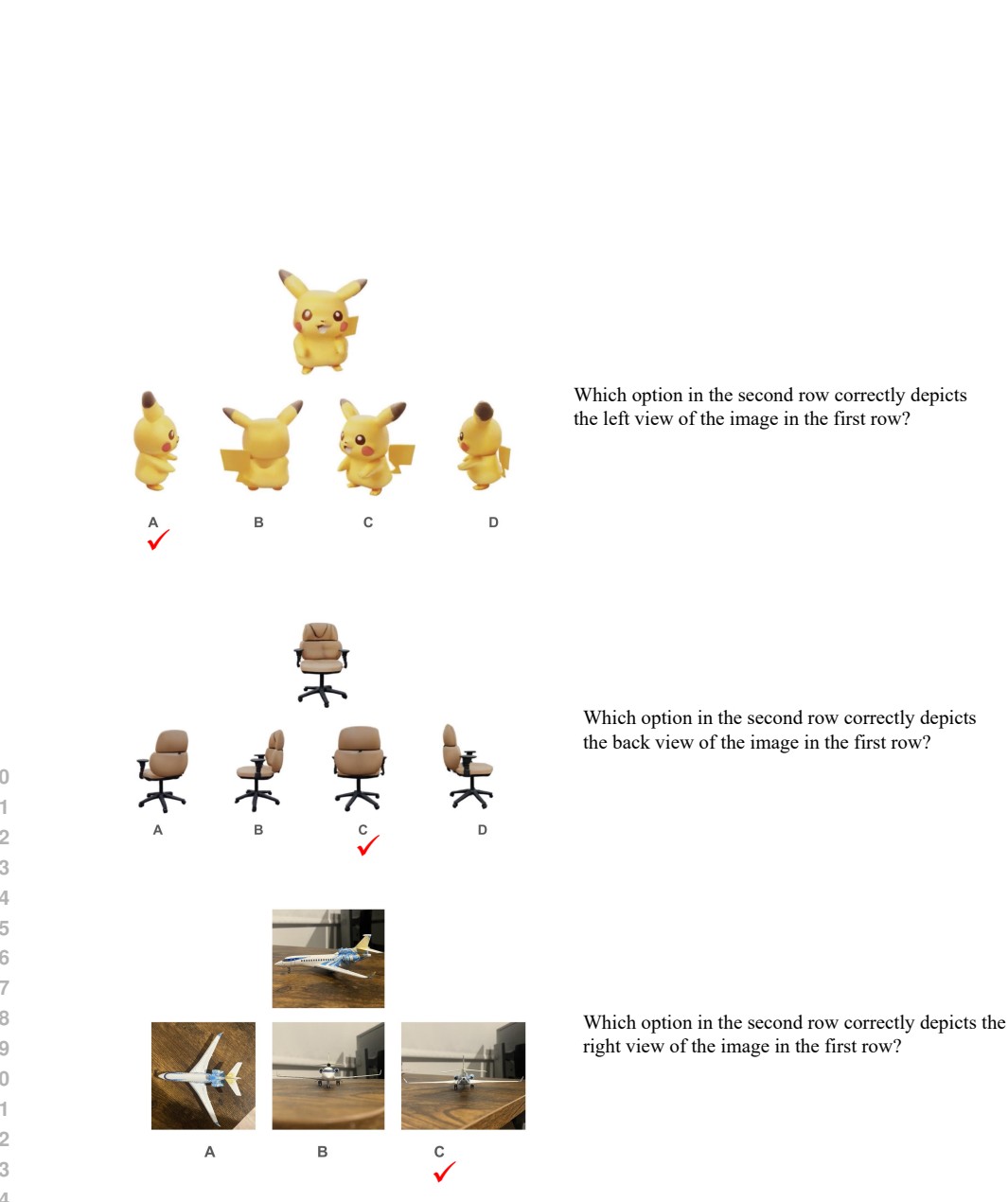

Which option in the second row correctly depicts the left view of the image in the first row?

Which option in the second row correctly depicts the back view of the image in the first row?

Which option in the second row correctly depicts the right view of the image in the first row?

Figure 5: Sample VRIQ natural 3D questions

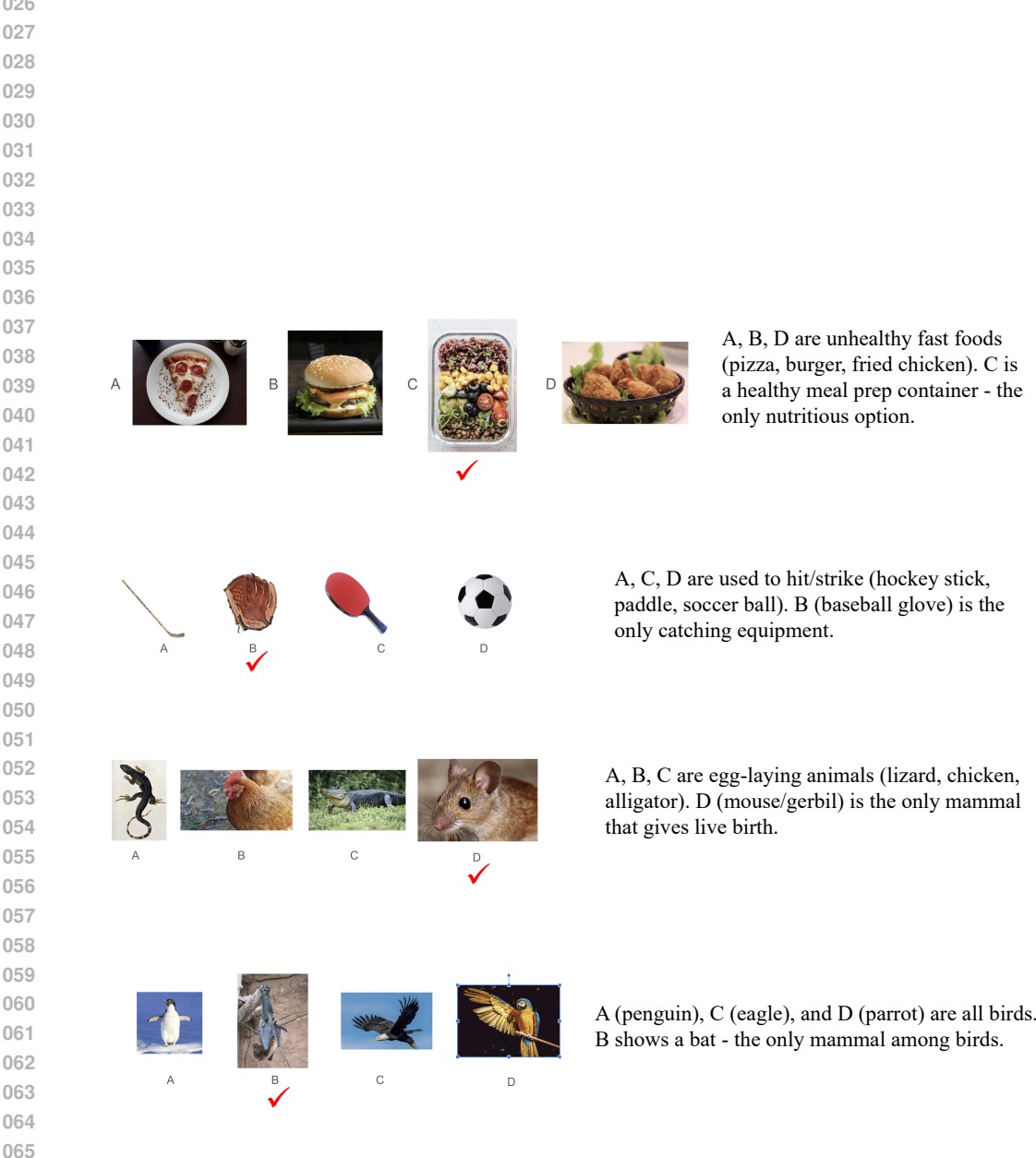

A, B, D are unhealthy fast foods (pizza, burger, fried chicken). C is a healthy meal prep container - the only nutritious option.

A, C, D are used to hit/strike (hockey stick, paddle, soccer ball). B (baseball glove) is the only catching equipment.

A, B, C are egg-laying animals (lizard, chicken, alligator). D (mouse/gerbil) is the only mammal that gives live birth.

A (penguin), C (eagle), and D (parrot) are all birds. B shows a bat - the only mammal among birds.

Figure 6: Sample VRIQ natural Odd One Out questions

