# OpenReview forum: "VRIQ: Benchmarking and Analyzing Visual-Reasoning IQ of VLMs"
_ICLR.cc/2026/Conference — ICLR 2026 Conference Desk Rejected Submission_

### Official Review · Reviewer_pJTp · 2025-10-25

**Soundness:** 1
**Presentation:** 3
**Contribution:** 2
**Rating:** 2
**Confidence:** 4

**Summary:**

This paper introduces VRIQ (Visual Reasoning IQ), a novel benchmark designed to evaluate the visual reasoning ability of Vision-Language Models (VLMs). The benchmark consists of 440 expert-authored items across five IQ test categories (Sequence Completion, 3D Visualization, Figure Rotation, Odd One Out, Matrix Prediction), each with both abstract puzzle-style and natural-image reasoning tasks to enable direct cross-domain comparison. The authors adopt a three-tier diagnostic evaluation framework: Tier 1 measures end-to-end accuracy, Tier 2 uses perceptual probes (isolating visual attribute extraction) and reasoning probes (text-only logical inference), and Tier 3 classifies failures into perception-only, reasoning-only, or combined types. Experimental results show that VLMs perform poorly on abstract tasks (average ~28% accuracy, near random) and moderately better on natural tasks (~45% accuracy). Tool-augmented models like GPT-o3 yield notable improvements but remain limited. Diagnostic analysis reveals that ~56% of failures stem from perception alone, ~43% from both perception and reasoning, and only ~1% from reasoning alone. Fine-grained probes further identify high failure rates in perception categories like counting (73.8%), position (58.2%), and rotation/orientation (52.4%). The paper concludes that current VLMs’ visual reasoning limitations are mainly due to perceptual bottlenecks, providing a diagnostic basis for model improvement.

**Strengths:**

1. Clarity: The paper has a logical flow—starting with the motivation of evaluating nonverbal reasoning in VLMs, followed by benchmark design, evaluation framework, experiments, and analysis.
2. Originality: The paper fills a gap in existing visual reasoning benchmarks by constructing parallel abstract-natural task families with identical logical structures, enabling controlled comparison of VLMs’ performance across symbolic and semantics-grounded reasoning. The hierarchical diagnostic probe framework (perceptual vs. reasoning isolation) is innovative, moving beyond monolithic accuracy evaluation to precise failure attribution.

**Weaknesses:**

- Limited benchmark scale: With only 440 total questions, the sample size is small for robust statistical analysis, especially when evaluating fine-grained perception categories. This may lead to unstable accuracy estimates, particularly for tasks with low model performance near random guess.
- Inadequate model coverage: The evaluated models lack diversity in two aspects: (1) No large-scale open-source VLMs (e.g., InternVL3-26B, Qwen2.5-VL-14B) are included, making it hard to assess the impact of model scale on visual reasoning; (2) Key SOTA proprietary models like Gemini-2.5-Pro and Seed1.5-VL are missing, my preliminary tests (w. examples in figure 1) with these models show high accuracy, which may weaken the generalizability of the conclusion that VLMs are unreliable abstract reasoners.
- Accuracy measurement vulnerability to randomness: Reporting raw accuracy for single-choice questions (with 4–5 options) is prone to randomness, especially when model performance is near the random baseline. Unlike benchmarks like MMBench [1] that use circular evaluation to mitigate this, the paper’s current metrics may overstate or understate true model capabilities, reducing conclusion credibility.
- Insufficient illustrative details for tool-augmented reasoning: Section 6.1 highlights that tool-augmented models (GPT-o3) achieve significant improvements, but no example traces of the tool-use process are provided. This makes it difficult to understand how tools (e.g., cropping, rotation) address perception bottlenecks, limiting the insight for developing tool-augmented VLM systems.

[1] MMBench: Is Your Multi-modal Model an All-around Player?

**Questions:**

- For Figure 1 (Sequence Completion - Natural Reasoning), the correct answer is unclear. Could you clarify the answer and the underlying logical rule (e.g., sequence of object attributes or transformations) to help verify task validity?
- In Table 2, the random guess accuracy for 3D V (3D Visualization) is 20%, but Figure 1 shows a 3D V example with 4 choices (suggesting 25% random guess). Please explain the discrepancy—are 3D V questions designed with 5 options, or is there another reason for the 20% baseline?
- In Line 420, you mention "QwenB" as one of the models for diagnostic analysis, but this model is not defined in Section 5 (Experimental Setup). Could you clarify whether this is a typo (e.g., Qwen2.5-VL-3B-Instruct) or an unmentioned model, and provide its details?

Please also help resolve concerns in the weakness section.

---

### Official Review · Reviewer_iAvs · 2025-10-27

**Soundness:** 2
**Presentation:** 2
**Contribution:** 2
**Rating:** 4
**Confidence:** 4

**Summary:**

This paper investigates the evaluation of visual reasoning capabilities in Visual Language Models and proposes the diagnostic benchmark VRIQ. Through a parallel task design of "abstract puzzles + natural image reasoning" and a three-tier evaluation framework (end-to-end accuracy - diagnostic probes - error classification), it systematically analyzes the performance and shortcomings of current VLMs in visual reasoning.

**Strengths:**

1. The paper is clearly written and organized.
2. The proposed three-tier evaluation framework decouples perception and reasoning as an integrated capability. It precisely attributes model failures to three categories: perception errors only, reasoning errors only and errors in both.
3. It simultaneously evaluates open-source and proprietary models (such as Qwen, GPT-4o, etc.), making it quite comprehensive. The result analysis is in-depth, with an appropriate combination of quantitative and qualitative approaches.

**Weaknesses:**

1. Although the authors claim that VRIQ is the first to conduct a parallel comparison in the abstract-natural dual domain, this setup is similar to MLLM IQ benchmarks such as MMIQ and MARVEL. It is hard to clearly demonstrate the unique advantages of VRIQ in terms of diagnostic accuracy or research inspiration.
2. The VRIQ benchmark contains a total of 440 questions, and the sample size of some reasoning categories is quite small. Small samples may lead to contingency in evaluation results.
3. For the tasks in the natural reasoning categories, if the scores are too high or relatively concentrated, this may mean that the benchmark lacks the ability to distinguish the capabilities between models.
4. For open-source models, only smaller ones have been tested. Models of 72B or larger with stronger capabilities have not been tested. For closed-source models, Claude and Gemini have not been tested.
5. Human reference performance should be provided to support the assertion that "humans can easily solve, but models fail".

**Questions:**

See weaknesses

---

### Official Review · Reviewer_eq1H · 2025-10-28

**Soundness:** 2
**Presentation:** 3
**Contribution:** 2
**Rating:** 2
**Confidence:** 4

**Summary:**

The paper proposes VRIQ, a diagnostic benchmark spanning five IQ-style categories across abstract and natural images, plus a two-tier probe methodology to attribute failures to perception vs. reasoning. Results suggest most errors are perceptual rather than reasoning.

**Strengths:**

The probe study is valuable. The perception-vs-reasoning decomposition and finer probe tags (count, position, rotation, 3D, etc.) are a useful analysis lens beyond aggregate accuracy.

**Weaknesses:**

- Small sample size. The core benchmark (440 items) limits statistical power. In addition, category counts are unbalanced between abstract and nature samples.

- The images are mix of open repositories and model-generated images, but sourcing, licenses are not described in the paper. Also how the images are generated using models are not described.

- The process for creating and validating data samples is not documented.

- Claims about “thinking with images” and tool use lack a transparent harness: was this an API implementation, what visual tools were allowed, how calls were budgeted/capped, decoding params, etc.

**Questions:**

- How the images are captured and how the final answers are validated? Please mention such important process in detail.

- How the o3 baseline model is implemented? What are the allowed tools?

- o3 is the only reasoning model being evaluated, it is worth to include other reasoning models, such as Gemini-2.5-pro, Claude-thinking modes, etc.

---

### Official Review · Reviewer_fxjD · 2025-10-31

**Soundness:** 2
**Presentation:** 3
**Contribution:** 2
**Rating:** 4
**Confidence:** 4

**Summary:**

This paper introduces a novel benchmark named VRIQ (Visual Reasoning IQ), designed to assess the visual reasoning abilities of Vision Language Models (VLMs). A core design principle of this benchmark is its "parallel domain construction," which includes both abstract, IQ-test-like puzzle tasks and natural image tasks that utilize the same reasoning logic (e.g., "odd one out" or "sequence completion"). The paper's main contribution is the introduction of a three-tier diagnostic framework that uses "Perceptual Probes" and "Reasoning Probes" to decouple the root causes of model failures.

**Strengths:**

1. As VLMs become more capable, it is crucial to discern whether they possess genuine reasoning abilities or merely rely on superficial statistical features from training data (shortcut learning). This paper directly addresses this core issue.

2. The conclusion that "the bottleneck is perception, not reasoning," while based on this specific dataset, is highly insightful. It points to a clear direction for future VLM improvements (i.e., strengthening foundational visual perception, especially spatial, counting, and 3D understanding).

**Weaknesses:**

1. The entire benchmark (VRIQ) contains only N=440 samples. In the fine-grained breakdown across 5 categories and 2 domains, aome sub-categories have an extremely small number of samples (e.g., Table 1 shows that each of the 5 "natural" domain categories has only 20 questions).

2. Due to the small sample size, the statistical reliability of the results is low. On a test set with only 20 questions, getting one or two more questions right or wrong causes a large fluctuation in accuracy (5%-10%). Therefore, conclusions drawn from such a small dataset (especially comparisons between models) are not robust.

3. The paper states VRIQ is "expert-authored" but provides no details on the creation process, quality control, or how data contamination was avoided (i.e., whether these IQ problems already exist online and might have been part of the VLMs' training data).

4. The paper defines "reasoning" as the ability tested by "text probes." However, this actually tests textual reasoning. A model might reason perfectly in text but fail to perform the same logical operation visually. Furthermore, some tasks classified as "perception" (like 3D visualization or counting) inherently involve complex reasoning. This binary split may oversimplify the problem.

**Questions:**

See above

---

### Note · Program_Chairs · 2026-01-17
**Submission Desk Rejected by Program Chairs**

The following references in this submission do not refer to real documents and/or have major errors in bibliographic information:

     Ziru Wang et al. Measuring and improving LLMs for math reasoning with visual contexts. arXiv preprint arXiv:2402.19427, 2024d.
    Zhuowan Li et al. MMCode: Evaluating multi-modal code understanding of llms. arXiv preprint arXiv:2406.17812, 2024c.